# Recent Advances in Polymer Additive Engineering for Diagnostic and Therapeutic Hydrogels

**DOI:** 10.3390/ijms23062955

**Published:** 2022-03-09

**Authors:** Sang-Wook Bae, Jiyun Kim, Sunghoon Kwon

**Affiliations:** 1Bio-MAX/N-Bio, Seoul National University, Daehak-dong, Gwanak-gu, Seoul 08826, Korea; sangwook.bae1@gmail.com; 2School of Materials Science and Engineering, Ulsan National Institute of Science and Technology, Ulsan 44919, Korea; 3Center for Multidimensional Programmable Matter, Ulsan 44919, Korea; 4Department of Electrical and Computer Engineering, Seoul National University, Daehak-dong, Gwanak-gu, Seoul 08826, Korea

**Keywords:** hydrogel, polymer additive, diagnosis, therapeutics

## Abstract

Hydrogels are hydrophilic polymer materials that provide a wide range of physicochemical properties as well as are highly biocompatible. Biomedical researchers are adapting these materials for the ever-increasing range of design options and potential applications in diagnostics and therapeutics. Along with innovative hydrogel polymer backbone developments, designing polymer additives for these backbones has been a major contributor to the field, especially for expanding the functionality spectrum of hydrogels. For the past decade, researchers invented numerous hydrogel functionalities that emerge from the rational incorporation of additives such as nucleic acids, proteins, cells, and inorganic nanomaterials. Cases of successful commercialization of such functional hydrogels are being reported, thus driving more translational research with hydrogels. Among the many hydrogels, here we reviewed recently reported functional hydrogels incorporated with polymer additives. We focused on those that have potential in translational medicine applications which range from diagnostic sensors as well as assay and drug screening to therapeutic actuators as well as drug delivery and implant. We discussed the growing trend of facile point-of-care diagnostics and integrated smart platforms. Additionally, special emphasis was given to emerging bioinformatics functionalities stemming from the information technology field, such as DNA data storage and anti-counterfeiting strategies. We anticipate that these translational purpose-driven polymer additive research studies will continue to advance the field of functional hydrogel engineering.

## 1. Introduction

By having a high water content, most hydrogels intrinsically have low mechanical strength with low stretchability (few times their original length) and low fracture energy (<100 J m^−2^), therefore they are difficult to shape in predesigned geometries [1]. However, potential benefits of applying hydrogels to fields such as medical soft robotics and wearable devices urged researchers to design hydrogels with enhanced physiochemical properties. Strategies to overcome the shortcomings involved (i) testing combinations of hydrogels from either natural (agarose, chitosan, collagen, hyaluronic acid, etc.) or synthetic (PEG, PVA, PLGA, PAAM, etc.) origins, (ii) testing crosslinking methods either physically (ionic, hydrogen bond, etc.) or chemically (covalent bond by radical polymerization, enzymatic crosslinking, etc.), (iii) adapting microfabrication tools for pattern design at various length scales, and (iv) introducing polymer additives (inorganic nanoparticles, graphene, carbon nanotubes, etc.).

Over time, pioneering studies introduced various advanced hydrogels including tough hydrogel [2,3,4,5,6,7], self-healing hydrogel [2,8], and injectable hydrogel [9,10]. Such advances in hydrogel polymer backbone engineering then, in turn, motivated the development of soft MEMS sensors and actuators, as well as drug screening and delivery platforms that were previously unattainable with conventional solid materials [11,12,13]. Additionally, in a relatively short period, we are now witnessing hydrogel products entering the clinic and the cosmetic field [14].

Although many microfabrication techniques are of great importance for hydrogel engineering [1], in this review, we focused on the various polymer additives that spurred numerous functionalities of hydrogel, especially in the biomedical field. We grouped these additives into four categories: nucleic acids, proteins, cells, and inorganic materials (Figure 1). We note that the term “inorganic” was used throughout this article to indicate materials not derived from biological cells rather than materials of non-carbon structure. Application-wise, we mainly focused on several key topics that are aimed toward translational medicine. Additionally, we highlighted rising inspirational applications of hydrogels inspired from cell biology (artificial cells and artificial microbiota) and computer science (anti-counterfeiting and data storage).

## 2. Nucleic Acid Additives

### 2.1. Nucleic Acid-Based Hydrogel Biosensors

Due to their unique stimuli-responsive characteristics and rich chemical modifications available, hydrogels have been widely accepted to design cheap and straightforward biosensors. These biosensors turned out to be facile alternatives to a number of sensors for detecting biochemical, pathogenic proteins or disease-specific genes [12]. Often, the reason behind this came from their unique analyte detection mechanism; nucleic acid probes incorporated into these gel-forming polymers can interact with target analytes and lead to macroscopic physical changes of the gel, such as swelling, color change, and viscosity change, which are easily detected with simple tools or even by the naked eye.

Nucleic acids, exemplified by DNA oligonucleotides, are highly programmable materials that have many modification techniques available to be integrated to numerous materials including semi-conductive materials and polymers. With these oligonucleotide probe-incorporated hydrogels, researchers developed various sensors for many types of oligonucleotide analytes, such as mRNA and viral RNA. One frequently used target analyte is microRNA (miRNA) because many miRNAs are known to be connected to the pathogenesis of diseases including cardiovascular diseases [15], nervous disorders [16], and cancers [17]. The aim is to analyze multiple miRNAs simultaneously using multiplex assays while maintaining sufficient specificity and sensitivity. In this regard, encoded microparticles with either shape [18] or graphical barcodes [19,20] proved highly effective and many initial proof-of-concept results were reported [21]. More recently, the topic is moving towards translational research for point-of-care (POC) device development. Efforts to increase compatibility with practical clinical samples [22] and detection sensitivity [23,24], and to integrate with microfluidics for robust readout [24,25] are being made (Figure 2a–d).

Other nucleic acid materials, such as aptamers and peptide nucleic acids (PNAs), are also used for analyte detection. Aptamer is a special type of nucleic acid probe because it binds not only with specific target analytes but also to their complementary sequences with high fidelity. Their binding targets span proteins, cells, and even inorganic materials, which makes aptamer attractive for applications in drug delivery, regenerative medicine, and molecular biosensing [28]. For example, pathogen-specific aptamers can be incorporated onto magnetic PEG microparticles encoded with reflection peak barcoding and used for multiplexed pathogen detection by simple mixing and magnetic separation [26] (Figure 2d). Aptamers against thrombin can be integrated on encoded particles for simple detection [29] or on pH-responsive hydrogel to create a microfluidic catch-and-release system [30]. Additionally, growth factor-specific aptamers can be placed on superporous hydrogels to sequester the target growth factor (e.g., PDGF) and perform programmable release upon que introduction [31].

### 2.2. DNA Hydrogels

An interesting crosswalk between hydrogel development and nucleic acid research is DNA hydrogel fabrication [32]. DNA hydrogels are hydrogels formed by the hybridization of DNA strands present in the gel-forming polymer network. Unlike other synthetic or natural biocompatible hydrogels, DNA hydrogels not only show good biocompatibility and biodegradability but also high programmability as seen in the phase transition (sol-gel, gel-sol, swelling, e.g.) occurring upon DNA sequence-specific triggers such as DNAzymes, aptamer target molecules, or antisense oligonucleotides. Moreover, diverse molecular recognition agents can be introduced into the polymer network [32]. Based on these characteristics, researchers have been exploiting new hydrogel-based smart sensors. DNA hydrogels are categorized into either pure DNA hydrogels formed purely by nucleic acid backbone [12,33,34,35] or hybrid hydrogels [27,36,37,38,39,40] composed of both nucleic acid chains and other gel-forming polymers.

Fabricating pure DNA hydrogels usually rely on Rolling Circle Amplification (RCA) [12,33,34] or DNA nanoassembly [35]. The forming of macroscale hydrogel can be detected by observing the rheological phase transition or enzyme-based colorimetric signal amplification [41]. Due to the recent COVID-19 pandemic, research for viral detection is gaining high appreciation. POC devices to detect pathogenic viral RNA usually rely on RT-PCR but recent developments showed that isothermal RCA-based DNA hydrogel formation can be applied for highly sensitive viral RNA detection as well [12,33]. In another study, they applied this to SARS-CoV-2 detection, demonstrating rapid diagnosis of 0.7 aM within 15 min [34]. Since RCA requires no thermal cycling, it shows great promise for future POC development.

Reported hybrid hydrogels were mainly used to utilize polyacrylamide, which is an acryl-based, biocompatible hydrogel material well-accepted in the biology field. The ease of incorporating acryl-modified nucleic acids into the polyacrylamide polymer network is one of the main reasons for this trend. Functionality demonstrations include logic gate-like programmable swelling by DNA strand displacement [36] and stress-responsive gel stiffness modulation [38]. miRNAs can also be detected electrically with ferrocene-tagged probes by measuring the current drop upon target miRNA-binding [40]. POC applications are also emerging (Figure 2e,f), such as the detection of Aflatoxin B1, Ochratoxin A, viruses, or glucose using target-specific aptamers and nanoparticle-based colorimetric quantification [27,37,42], or quantum dot (QD)-based fluorescence quenching [43]. The cascade reaction of G-quadruplex formation, hemin-based H_2_O_2_ formation, and monomer crosslinking were utilized for detecting ATP [44].

### 2.3. Injectable Cancer Immunotherapy Drugs

Cancer immunotherapy drugs comprise molecular (e.g., therapeutic vaccine and checkpoint inhibitor) and cellular (e.g., CAR-T cell, ex vivo T, NK, and APC cells) therapeutic agents engineered to educate the body’s immune reaction against tumor antigens [45,46,47]. Among those, cancer vaccines are usually composed of DNA vectors or RNA strands encoding cancer-specific antigens. After injection by various routes (intravenous, intramuscular, subcutaneous, or intracutaneous), these agents transfect into cells. Additionally, through subsequent antigen-processing pathways of antigen-presenting cells (APCs), tumor antigens are imprinted in the immune memory as toxic, thus inducing anti-tumor immune reaction throughout the body. The intriguing aspect is that although these vaccines are locally delivered to reduce systemic adverse effects, their therapeutic effects are still systemic due to the adaptive immune mechanism called “abscopal effect” [48].

Nucleotide-based vaccines have a number of benefits over native protein antigen vaccines, such as being relatively cheap to produce, thermally stable, and highly programmable in design. However, they tend to produce weak immunogenicity (i.e., the ability to induce immune reaction against given antigen) due to intrinsically low cell transfection efficiency. Moreover, the current necessity of systemic high dosage is leading to an increase in cost and occasional immune-related adverse events (IRAEs) [49].

In this respect, enabling local injection of these DNA vaccines with prolonged localized high concentration would be desirable. Thus, recently, new tools using injectable smart hydrogels have been introduced to cancer vaccine development [50,51,52]. These injectable hydrogels can encapsulate and locally release macromolecular DNA vectors as well as small molecule adjuvants (e.g., CpG and MPL). This is anticipated to allow for effective immunotherapy at lower doses with reduced adverse effects. Injectable hydrogels also show other good qualities such as facile formulation, surgery-free administration, and shape fitness to body cavities. So far, polysaccharides (HA, alginate, chitosan, etc.) collagen, PEO-PPO, and PEG-PLGA constitute the majority of the reported injectable hydrogels because of their verified biocompatibility and/or biodegradability. However, further material inventions are required to enable effective deep-tumor injection and controllable cargo release.

In a report, as a cancer therapy demonstration, poly (ε-caprolactone-co-lactide) ester-functionalized hyaluronic acid (HA-PCLA) embedded with OVA expressing plasmid and GM-CSF was locally injected to a murine melanoma (B16) expressing OVA [52] (Figure 3a,b). The PCLA-based copolymers exhibited sol-gel phase transition between room and body temperature, thus enabling hydrogel injection. It also exhibited good biodegradability compared to other polymers such as pluronic, poly-phosphazane, or PNIPAM. The adjuvant GM-CSF enhanced the local accumulation of APCs and uptake of the OVA protein, eventually leading to an enhanced activation of OVA-specific CD8+ T cells and their tumor cytotoxicity.

Another usage of nucleotide for cancer therapy is using DNA aptamers that specifically block checkpoint markers on immune cells. Checkpoint inhibitor-based T cell reactivation is at the forefront of cancer immunotherapy and usually uses checkpoint-specific antibodies. Replacing antibody with DNA aptamer has several advantages such as higher thermal stability, lower immunogenicity, and, most importantly, deeper tumor penetration due to its smaller size [53]. However, after tumor penetration, its retention rate has to be enhanced with additional tools for sufficient therapeutic efficacy. A recent study realized long-term PD-1 aptamer retention in the tumor site using polyaptamer hydrogels [54] (Figure 3c,d).

**Figure 3 ijms-23-02955-f003:**
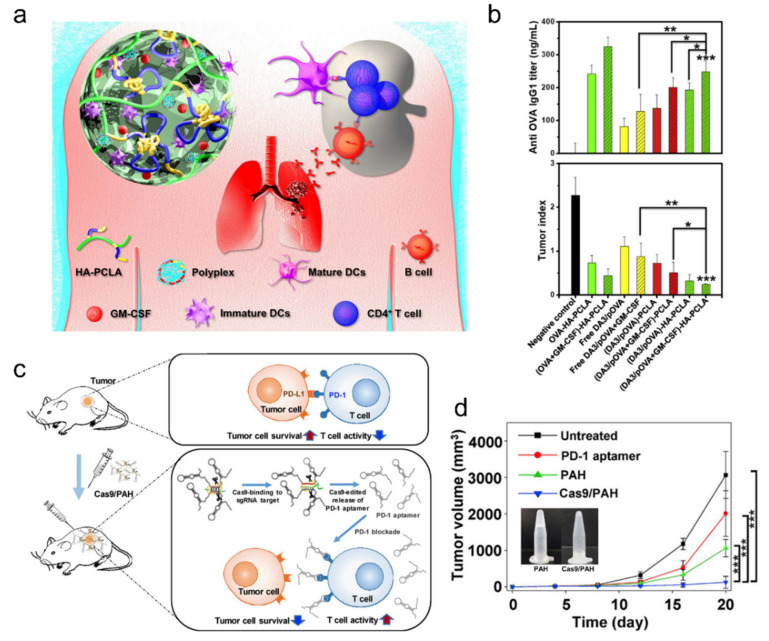
Hydrogel drug delivery platforms with nucleic acid additives. (**a**,**b**) Injectable cancer DNA vaccine hydrogel. (**b**) Demonstration of humoral anti-OVA immune response (production of anti-OVA IgG1 antibody) in BALB/c mice induced by various injected DNA polyplex OVA; adapted with permission from [52]. Copyright 2020 Elsevier. (**c**,**d**) Anti-PD-1 aptamer delivery using injectable polyaptamer hydrogel; adapted with permission from [54]. Copyright 2019 Elsevier. Cas9 induces polyaptamer hydrogel dissociation and tumor local blockade of PD-1 checkpoint on T cells, thus inducing higher antitumor effect (**d**).

### 2.4. Recent Highlight: DNA Data Storage

DNA data storage is a recently initiated exploration using DNA oligonucleotides as direct data storage material, an alternative to semiconductors which are expected to be depleted in 2040 if the currently increasing demand on digital data storage continues [55]. DNA data storage is based on the conversion of digital binary (0 or 1) data to chemically synthesized quaternary (A, C, G, or T) data. It allows for the highest physical information density of petabytes of data per gram, the durability of centuries, and no energy being required for data preservation. The field grew rapidly with the emergence of high-throughput DNA reading (i.e., next generation sequencing (NGS)) and advances in DNA writing (chemical oligonucleotide synthesis) technology. Most recently, researchers are focusing on reducing the reading error rate of stored data and reducing the data storage cost through computational algorithms and chemical synthesis techniques [56,57].

However, for the next step of industrial application of DNA data storage, Choi and colleagues developed a DNA micro-disk fabricated by incorporating data storing DNA material on QR-encoded hydrogel microparticles [58] (Figure 4). These QR-coded microparticles work as a data management system for access. They demonstrated that these micro-disks could work as a write-once-read-many (WORM) memory. They showed that the multiple file library, which included maps and texts, can be stored into primer sets, stored in a DNA micro-disk, and then recovered selectively through PCR amplification against the primer set designated for the targeted file upon demand.

Such adaptation of IT technology to engineer functional hydrogels is one of the most recently conceptualized, cutting edge trends in polymer design. As we are in the era where information is the most highly valued asset, we expect that such DNA data storage technologies will find many needs not only in the biomedical field for storing health-related data but also in the wider community of science, finance, education, and any other industry involving IT.

## 3. Protein Additives

### 3.1. Antibody-Laden Hydrogels for Multiplexed Immunoassays

Among the many kinds of protein, antibody is probably the most widely used additive for hydrogel sensors. One notable use of antibody-incorporated hydrogels is microparticle-based immunoassay [59,60,61,62]. Such assays are one of the key applications of encoded microparticles. Among the various methods of fabricating hydrogel microparticles, such as batch emulsion, microfluidic emulsion, lithography, and electrodynamic spraying [63], lithography is advantageous in creating hydrogel microparticles for multiplexed immunoassays because it uses masks or molds that can tightly control geometrical features, thereby enabling microscale graphical encoding on microparticles.

The key element of such multiplex immunoassays is fabricating encoded microparticles with functional antibodies incorporated on the surface. Using polyethylene glycol diacrylate (PEGDA) mixed with photo-sensitive initiators exposed under patterned UV light has been the most popular way of fabricating these microparticles [59]. The reasons for this include the low material cost, good biocompatibility, and good polymer chain length tunability for physiochemical property design. An early method of incorporating antibodies to these microparticles was copolymerization, which involves mixing antibodies and monomers before photolithograpy. Albeit simple, it suffers from low surface antibody loading density and antibody aggregation, which is aggravated by an increase in the antibody concentration [62]. However, more recently, facile chemical incorporation of antibodies’ post-particle synthesis became available. This method uses either chemical linkers such as heterobifunctional PEG (Thiol- PEG-NHS) [62] or amide bond formation [64]. Starting from the early demonstrations of the 3~4 multiplexing capability and three-log dynamic range with 1 pg/mL detection sensitivity [65], the topic evolved rapidly, especially in regarding its multiplexing capacity. Nowadays, commercial Luminex multiplex immunoassays are capable of around 100 plex capability with around a 1 pg/mL limit of detection for cytokine assays [66].

However, when regarding the entire assay scheme, there are still many steps that require technical improvement. For example, analyzing these encoded particles has mostly been based on optical microscopy, which, even when using the non-fluorescent colorimetric method, is complex and expensive for POC settings. In this regard, a cheap scanner-based multiplex colorimetric immunoassay was recently developed [67] (Figure 5a,b). The authors showed their uniquely encoded particles that allowed for scanner-based colorimetrical autoantibody detection and the potential multiplexing capacity of two million. Another compelling research topic is improving strategies for analyte sampling from various sources. One good example is preparing analyte samples (cytokine, microbiota, etc.) from skin, which is notoriously difficult, yet technical improvement requisitions from the cosmetic field is ever increasing. A recent report deals with this problem by assembling antibody-incorporated microparticles into patch-like array and applying it directly onto skin for multiplexed cytokine measurements on skin [68] (Figure 5c,d). Although the technique requires fluorescence microscopy at this point, it is plausible to integrate facile assay readout schemes for better adaptability for POC and cosmetic applications.

### 3.2. Peptide/Enzyme-Laden Hydrogels for Bioactive Implants

Bioactive peptides (e.g., RGD) and enzymes are a class of proteins frequently combined with hydrogel for biomedical applications. These protein-laden hydrogels are usually designated as bioactive hydrogels and typically formed by mixing enzymes or peptides with macromolecular polysaccharides, proteins, and peptides. Often, this leads to in situ enzymatic crosslinking, polymerization, or compartmentalization of the resulting hydrogel. These bioactive hydrogels could be used for in situ wound protection and health monitoring, or as therapeutic agents [69,70,71,72]. One good example is the use of microporous annealed microparticles (MAPs) for easy in vivo injection and effective wound healing (Figure 6). It was shown that with proper MAP design, it could also elicit adaptive immune response for boosted wound healing and the regrowth of hair follicles.

Although many enzyme-laden bioactive hydrogels were introduced, glucose oxidase (GOx) is the most widely adapted enzyme for such applications. Either by itself [73] or synergistically with horseradish peroxidase (HRP), it could be used in situ to form polymer networks at room temperature [74,75] or to realize 3D printing. When these polymerization schemes were applied to supramolecular hydrogels made from hydrogelators [76], supramolecular peptides [77], or low molecular-weight gelators (LMWGs) [74,78], the resulting hydrogels presented self healing capabilities. When N-hydroxyimide-modified silica NPs were introduced, GOx formed stretchable, tough hydrogel networks at anaerobic conditions and were applicable for bio-scaffold and cell culture [79]. Other reports showed that enzyme-laden hydrogels could be used for cell type-specific drug delivery [80], tumor killing [81], tumor-specific in vivo fluorescence imaging [82], or pH-sensitive drug delivery [83].

### 3.3. Hydrogel-Based Artificial Cells

An eccentric and rather recent topic regarding hydrogel is artificial (or synthetic) cells. Artificial cells are cell-mimetics with similar microscopic scales (10~20 um in diameter) that can be formed from liposomes, coacervates, or hydrogel droplets. The goal is to synthetically produce cell-like functionalities for potential applications in drug delivery, drug encapsulation, and regenerative medicine [84]. Although currently perceived as rather radical, some researchers even envision building artificial intelligence or embodied cognition in a circuit of artificial cells [85]. They predicted that this could be achieved by the synergy of artificial intelligence technology and artificial cell technology through the ‘understanding-by-building’ approach. Such synergy between molecular and computational designs would require accurate modeling of the chemical/biological reaction dynamics involved in most biological circuits. Overall, many challenges lie ahead for developing artificial cells that possess compelling biological functionalities.

Hydrogel engineering techniques may be useful for tackling some issues involved in creating artificial cells, such as the compartmentalization issue. In natural cells, to successfully perform the numerous functionalities of biological life, they adapted highly compartmentalized architecture through either membrane formation (mitochondria, lysosome, etc.) or non-membrane-bound mechanisms (nucleolus, P-bodies, etc.) [86]. How membrane-bound organelles are formed by lipid vesicles and membrane proteins has been well studied. However, how non-membrane-bound organelles form remained a mystery until recent years. Recent studies revealed that several scaffold proteins trigger biomolecular condensate formation through phase-separation, a phenomenon that was already well recognized in the polymer engineering field [87,88,89,90]. Unlike membrane-bound organelles, these biomolecular condensates are sensitive to environmental and compositional changes, which frequently occur in the cell through post-translational modification and other molecular dynamics. Thus, harnessing these scaffold proteins could enable compartmentalized biochemistry within an artificial cell in a spatially and temporally controllable manner. Although currently most artificial formation of condensate hydrogels is performed in living biological cells [91,92,93], artificial cell-based approaches using polymer–peptide hybrids are also emerging [94] (Figure 7a,b).

Although there are many reports about fabricating artificial cells, the majority is liposome-based artificial cells and most hydrogel-based artificial cells, in fact, have more similarities to smart biosensors and actuators than to biological cells. Nevertheless, a number of hydrogel-based artificial cells showing compelling cell-like functionalities have been introduced. Such cells include artificial antigen presenting cells [95] (Figure 7c,d), which are ellipsoidal PVA hydrogel particles conjugated with T cells stimulating anti-CD28 antibody and the MHC-antigen complex. These ellipsoidal artificial APCs induced higher T cell stimulation than spherical nanoparticles due to its larger cell contact area. Another good example is the demonstration of artificial cell-cell communication [96] (Figure 7e,f) where an artificial enzyme that mimics cytochrome P450 was used to facilitate signal transfer between artificial cells and mammalian HepG2 cells, thus connecting the artificial cell and biological realm.

### 3.4. Systems for Detecting Pathogenic Cells and Molecular Diagnostics

Recently, hydrogel-based cell-capture platforms are emerging as the needs from the pharmaceutical and clinical field are growing. Unlike basic cell sensors for quantifying target cells, cell-capturing platforms are, in many cases, developed to enable downstream molecular analysis of the captured cells. The necessity for such a platform usually is in regard to the fact that the target cells are rare yet contain valuable clinical information for either diagnostic (e.g., circulating tumor cell and CTC) [97] or therapeutic (e.g., anti-antigen antibody lead) [98,99] purposes. Coupling cell capture with downstream molecular analysis might seem straightforward but many design restrains emerge because the viability and molecular state of the captured cell have to be intact or, at least, fairly preserved. Materials for cell manipulation, capture, wash, and/or release have to be biocompatible not only at the cellular level but at the biochemical level because most downstream molecular analyses involve using environment-sensitive proteins such as enzymes and antibodies. Optimizing such a platform therefore requires avoiding the use of cytotoxic monomers, initiators, detergents, oils, or, in some extreme cases, single-cell sequencing, which is even widely used in cell-staining dyes. Thus, it is only natural to use hydrogel and biochemically compatible additives as the building blocks of such a platform.

In one report, researchers developed a CTC-capturing microfluidic chip that is also capable of selectively retrieving and analyzing the captured CTCs with the aid of a pulsed infrared laser system [100] (Figure 8a–c). The cell-capturing pillars in the fluidic chip were composed of biocompatible, CTC-capturing antibody-conjugated polymer pillars. After selective CTC capture from blood samples, pillars coated with CTCs were selectively retrieved by pulsed lasers into a biochemical reaction mixture for downstream genome analysis. The researchers first analyzed the whole genome of the single CTCs and, in a follow-up study, further improved the microfluidic system to enable gene expression analysis in situ [101] (Figure 8d,e). Scaling up these cell-capturing techniques to a large cell library using screening platforms is also being exploited. As shown in one report, a high-throughput phage-display system was shown to be capable of analyzing 14% of a single-chain variable fragment (scFv) library of 5 × 10^5^ complexity and positively selected 78 antigen-reactive scFv as drug candidates. Compared to conventional bio-panning, the technique required fewer panning rounds, thus providing broader screening capabilities for more potential drug candidates [102]. Due to the recent COVID-19 pandemic, collecting or detecting infectious virions with high sensitivity became an important aspect for diagnostic tools [103]. Barclay and colleagues showed that hydrogel particle-based virus concentration significantly improves such detection sensitivity compared to conventional RT-PCR without virus concentration [104].

## 4. Cell Additives

### 4.1. Mammalian-Cell Laden Hydrogels for Tissue Engineering

When it comes to incorporating living cells inside a 3D matrix for cell growth, using hydrogel becomes almost indispensable. From simple cell encapsulation to complex tissue engineering, hydrogel scaffolds are known to be the best choice to tailor the cells’ native environment. Characteristic properties of hydrogels make them especially appealing for repairing and regenerating soft tissues and organs. Solid free-form fabrication (SFF), which is a class of rapid prototyping based on laser (SLA, 2PP), nozzle (FDM), or printer (Inkjet, 3DP) mechanisms, was successfully adapted for printing these hydrogel materials into 3D scaffolds for tissue engineering. These printers use hydrogel inks composed of either synthetic polymers (polyethylene glycol, poly lactic-co-glycolic acid, polyvinyl alcohol, etc.) or natural polymers (hyaluronic acid, agarose, gelatin, chitosan, alginate, collagen, etc.). The field has rapidly matured over the past decade with extensive archives about the aspects of each printing method and printing materials available [105,106,107]. Additionally, with the aid of clinical imaging techniques, such as computed tomography (CT) and magnetic resonance imaging (MRI), it became rationally possible to produce reconstructive tissues or organs customized for patients [108].

Although there are many tissue types targeted for artificial fabrication, the most widely studied tissue types for 3D-printed hydrogels are skin, cartilage, cardiac, and neural tissues. For skin tissues, cell types, such as fibroblasts and keratinocytes, are being used for bio-inks directly printed on the wound site [109,110]. Cartilage tissue cells, such as chondrocytes, can be mixed with hyaluronic acid, one of the major components of cartilage tissue, and other materials (i.e., PLA) to form printable bio-inks to form mechanically enhanced structures [111,112,113]. For cardiac tissues, many cell types, including vascular endothelial cells, cardiomyocytes, and smooth muscle cells, are either seeded on pre-printed hydrogel scaffolds or printed simultaneously [114,115,116]. A recent study advanced this particular field to the point where collagen-based 3D printing of whole human heart is now possible (Figure 9a,b). As for neural tissue fabrication, it was shown that human cortical neural stem cells can be grown in a matrix consisting of alginate, carboxymethyl chitosan, or agarose [117]. The resulting tissue showed effective neural cell-like GABA expression after 10 days of in vitro proliferation. In another example, a bio-ink made of mouse neural stem cells embedded in PCL was used for generating aligned fibers for guided cell growth and lengthened neurites by stereolithography and electrospinning [118].

Adapting functional (or dynamic) hydrogels for scaffold design provides many advantages over static hydrogels. Creating self-healing hydrogel scaffolds is probably the most widely studied subject. Such self-healing at biologically relevant conditions can be realized with self-assembled low molecular weight gelators (LMWGs) [122], self-assembling peptides [123,124], hyperbranched polymer cross-linking [120], supramolecular crosslinking [119] or host-guest complexes [125]. Such hydrogels usually self-heal and/or display shear-thinning and reassembling capacity, which makes them suitable candidates for developing injectable inks (Figure 9c,d). Stimuli-responsive hydrogels respond to external stimuli and undergo structural or chemical changes. Among the many kinds, thermal-responsive hydrogels [126,127], enzymatically degradable hydrogels [128], and spatiotemporally photo-cleavable or crosslinked hydrogels [121,129] are the most well studied (Figure 9e,f).

Although there are many obstacles to developing new printable biomaterials, the rational designing of scaffolds based on hydrogels is turning out to be a good strategy. In general, printable biomaterials must have good biocompatibility and mechanical properties, and should support printing complex internal structures (voids, tortuous paths, etc.). These are necessary characteristics to enable cell infiltration, normal cell growth, and, when implanted, successful integration with the native tissue environment [130]. Meeting these criteria involves understanding complex cell-cell and cell-matrix interactions in natural tissues and tailoring these environments with proper polymer structures. This means that pairing cells with the right polymer matrix is important and recent molecular cell biology studies of cell-hydrogel matrix interaction mechanisms are providing critical insight in this respect [128,131,132].

It is unlikely that there is a universal polymer design fit for all tissue types. Optimal design will vary depending on the application as well as tissue type. Even for a particular application, it is rare to see one polymer material providing all the required ink properties. Designing scaffolds with composite hydrogels, block copolymers, or an interpenetrating network (IPN) composed of multiple types of polymers are therefore the usual strategies. However, currently, researchers are confronted with trade-offs and optimization limits due to the rather short list of validated polymers for tissue engineering [105]. Such design constraints could be alleviated by creating novel polymers that tailor native tissue and are applicable to tissue engineering procedures. Additionally, further improvements of 3D cell printing techniques are required because they currently suffer from either low resolution (FDM and bioplotting), low mechanical strength of the printed object (bioplotting), or low compatibility with live cell printing (SLS and SLM).

### 4.2. Microorganism-Laden Hydrogel-Based Synthetic Microbiota

A scientific understanding of the microbiota in nature has great value for both therapeutic and industrial purposes. Harnessing the molecular properties of microorganisms has already enabled many industrial productions of the chemicals and drugs we now take for granted. Inspirations from material engineering led to the invention of engineered living materials (ELMs), which are materials incorporated with living cells. These ELMs can be utilized to either generate exotic properties (self-healing concrete, self-assembly-based mass production, etc.) or provide alternative sources for natural materials (leather, wood, etc.) [133]. In nature, different strains or species of bacteria frequently mix together and form packs (i.e., microbiota), leading to complex chemical interactions which could build up to smart phenotypes such as a resistance to antibiotics, formation of biofilms, and displaying of evolutional fitness in the host environment [134]. Deciphering these phenotype mechanisms is important to understand and treat human microbiota-related diseases. To understand these complex molecular and cellular mechanisms in vitro, one must create complex synthetic environments that sufficiently tailor to that nature.

Synthetic microbial consortia, a topic that emerged from interdisciplinary works among the microfluidics, material engineering, and microbiology communities, comprises in vitro platforms for studying the complex interplay within an artificial microbial consortium mimicking natural environments such as air, water, soil, plants, and animal bodies [135]. The final goal is to produce systemic platforms to aid scientific microbial evolution studies and industrial bacteria-based production of food and drugs. Studies revealed that the ability to spatially organize microbial populations is important for understanding microbial communication. To create spatially organized environments for microbial populations, it is rational to use biocompatible and spatially organized hydrogel materials. As one example, researchers developed a system combining micro-3D printing and scanning electrochemical microscopy (SECM) to analyze the chemical communication of spatially structured microbial communities [136]. They created gelatin microtraps for each bacterial aggregate and positioned ultramicroelectrodes right above the microtrap roof to measure current permeability changes by the bacterial metabolites. This tool enabled studying the minimal signal threshold for the quorum-sensing of pyocyanin among different strains of Pseudomonas aeruginosa.

Such understandings of cell-cell communication are now being harnessed for industrial purposes, as shown in more recent studies where hydrogel-laden microbes for on-demand production of chemicals [137] or stem cell differentiation [138] are being demonstrated (Figure 10a–c). In another study, researchers used chitosan-coated alginate-based bacteria-laden microfibers to fabricate a 3D microbiome-within-a-membrane model [139] (Figure 10d). The microfibers spatially separated constituent bacterial species while mucin and caboxymethyl cellulose (CMC) were supplemented to mimic biofilm formation on the intestinal surface. They designed hydrogel fibers holding different bacteria strains to create a contact-independent co-culture. With these, they devised a polymicrobial intestinal lumen model to demonstrate contact-independent microbial crosstalk and observed its impact on the intestinal epithelium. These bacteria-laden hydrogel structures showed that microbiota-like functionalities can be artificially fabricated and harnessed for the production of various biomaterials. Although these technologies are at the initial proof-of-concept stage, further studies using multiple strains or microbiota obtained from natural origins will help gain further insights on developing better artificial microbiota platforms.

### 4.3. Microorganism-Laden Hydrogels for Protein and Antibiotic Drug Screening Platforms

The abovementioned tissue engineering and in vivo cell delivery applications usually deal with mammalian cells but microorganisms such as virus and bacteria can also be incorporated into hydrogels. The field most explored using such microorganism-laden hydrogels is in vitro cell assay for drug screening and disease mechanism studies. In vitro assays that utilize mammalian cells, organoids, and 3D-printed organ models are well reviewed in other articles [140,141]. Here, we focused on recent developments in microorganism-based in vitro assays.

Although many viruses are pathogenic entities, they are now the key source of many important biotechnology tools. Using these as vectors is one of the most widely used ways to transfer genes to cells in vitro and in vivo. A study showed that by incorporating these viral vectors to hydrogel microparticles, one could generate viral patch libraries for multiplex gene delivery [142] (Figure 11a). The virus-laden shape-coded hydrogel particles, with each shape with different adenoviral vectors for different kinds of GPCR protein, were placed directly onto a monolayer of cultured cells. This way, they were able to perform localized multiplex gene delivery within a single cell sample without cross-reaction among different viral vectors. The technique was then improved by a shape-by-shape assembly-based gene delivery technique to further increase the delivery combinations [143] (Figure 11b).

Phage display is a well-known high-throughput screening platform for selecting antibodies that bind to targeted antigens. However, conventional protocol requires a long incubation time for large colony formation and the sampling process of these colonies are extremely laborious. To overcome this, a research group developed TrueRepertoire^TM^, a rapid antibody screening platform based on agarose droplet array [145]. For this, a micropillar array chip was first incubated with a mixture containing liquid agarose and a diluted antibody library-transformed *E. coli* sample. This lead to the formation of agarose droplets around each pillar with one or no *E. coli* cells per droplet. After incubation for colony growth, they selectively retrieved agarose droplets that contain an E.coli microcolony. With this technique, they were able to select approximately 1000 unique scFv-targeting hHGF while maintaining the heavy-chain and light-chain pair information.

The antibiotic susceptibility test (AST) is a clinical test for potentially septic patients to test pathogenic bacterial infection and select proper antibiotics for prescription. However, conventional AST takes more than a day due to the overnight (16~24 h) bacterial growth incubation required. Choi and colleagues developed a rapid AST that significantly reduces the assay time required for determining the minimum inhibitory concentration (MIC) to 3~4 h [146]. Such performance relies on their single-cell imaging technique. In their microfluidic chip, bacteria cells are fixed in an agarose matrix which interfaces with another fluidic channel filled with antibiotics. This way, the proliferation rate of each fixed cell can be monitored in the presence of antibiotics diffused into the agarose channel. By preparing multiple replicates of such a chip with varying types and concentrations of antibiotics, they were able to construct an AST platform that produces similar MIC profiles to the CLSI standard but significantly faster than conventional optical density (OD)-based tests.

In a follow-up study, they created a novel single-cell morphological analysis (SCMA) technique-based AST [147] (Figure 11c–e). In this study, they shared their insight that not all bacteria respond to susceptible antibiotics by reduced proliferation rate (i.e., decrease in occupying area). In other words, some bacteria (such as MRSA) proliferate even when exposed to their susceptible antibiotics but with an altered morphological pattern (e.g., filamentous growth). They tested their SCMA technique using standard strains and various clinical samples which resulted in a fast (4 h) AST turnaround time with high accuracy that satisfies U.S. FDA recommendations. High accuracy was achieved even for challenging strains such as antibiotic-resistant strains and filament forming β-lactamase-positive strains. It was also shown to be applicable to extremely slow-growing Tuberculosis (TB) strains [148]. Recently, the researchers developed a rapid AST platform that can perform tests directly with whole-blood samples, thus removing the lengthy blood culture required for previous AST and reducing the total AST test duration from 60 h to less than 24 h. These platforms are now being commercialized for routine hospital tests by a company called Quantamatrix and are expected to greatly benefit urgent infectious patients.

## 5. Inorganic Additives

In general, hydrogels incorporated with inorganic nanoparticle (NP) additives are categorized as nanocomposite hydrogels [144]. NPs can be formed with various materials such as polymers, minerals, metals, and semiconductors. Incorporating these NPs to hydrogels led to numerous creations of nanocomposite hydrogels with enhanced physiochemical properties or new functionalities. For example, mineral NPs, such as silicate mineral clays, enhanced the tensile strength of PVA hydrogels [149]. Polymeric dendrimer NPs have been used as micelle to perform drug encapsulation and delivery [149,150,151]. Metallic AuNPs have useful electronic and optical properties which are being exploited for developing biosensors [152] while AgNPs display antibacterial properties [153]. Magnetic NPs have applications for MRI [154], remote heat induction, and drug release [155]. Nanocomposite hydrogels with stimuli-responsive properties could also be harnessed as smart materials with applications including soft robotics and diagnostic/therapeutic nanomedicine.

### 5.1. Chemical Reaction-Screening Platform

The usual applications for synthesizing hydrogel materials containing chemical drugs are either drug encapsulation or in vivo drug delivery. Stimuli-responsive hydrogels allow for spatiotemporal control of drug delivery, which is preferred for targeted and/or long-term treatments. They also enable theranostic applications, performing both diagnosis and therapeutic function with a single agent [154].

However, a number of recent reports demonstrated a distinctive functionality: high-throughput chemical reaction-screening. Compared to conventional high-throughput drug-screening systems, microparticle-based drug delivery systems are capable of heterogeneous drug delivery to individual micro-reaction chambers at low cost and labor input [155]. Additionally, to construct a chemical reaction-screening platform with these microparticles, one could imagine creating an encoded microparticle-based drug library with each particle loaded with a specific drug (or drug candidate) assigned to its code. However, one practical hurdle to create such a library was precisely controlling the dosage embedded in each particle, which was difficult with conventional absorption-by-drying. Researchers discovered that freeze-drying of the drug-microparticle mixture leads to significant improvement of uniform drug loading compared to the normal drying method [156] (Figure 12a,b). The process proved to be valid with anti-cancer drugs such as Doxorubicin and Erlotinib, with a clear linear correlation between the loading concentrations and release concentrations.

Based on these drug-loaded hydrogel microparticles, researchers were able to create a one-step pipetting-based drug-screening platform that systemically determines the optimum anti-tumor drug combination [157,158] (Figure 12c,d). The developers emphasized that even with the same drug combination, the cytotoxic effect can vary by the treatment sequence and that their platform is able to identify which combination, dosage, and sequence has the best anti-cancer effect. They also noted that their platform can replace thousands of pipetting to a single pipetting step, which could potentially lead to a significant decrease in workload and cost in hospitals and laboratories.

### 5.2. Soft Robotics

Soft robots, which converts external energy to mechanical work, exhibit flexible motions tailored to living organisms. Notably, hydrogels, among many solid materials, have one of the widest ranges of applicable actuation stimuli. Additionally, hydrogel-based soft robots have, in many medical situations, better compatibility with the human body than conventional rigid material robots. Thus, despite there being other soft materials available, hydrogels are becoming almost indispensable for developing soft robots for in vivo applications such as drug delivery robots and artificial organs [159,160].

Like other smart material-based tools, hydrogel soft robots use physical or chemical stimuli for actuation. Hydrogel actuators are typically recognized as sluggish because their motion relies on diffusion, mass transportation for their phase transition, and skin layers which delay the bulk shrinking or swelling. However, new actuation mechanisms are continuously being developed to overcome these problems.

The three most commonly used actuation stimuli are thermal, electrical, and magnetic stimuli. For thermo-responsive hydrogels, exemplified by PNIPAAm, the main role of nanocomposite additives is photo-thermal conversion (photo-stimulation) or thermal conduction (temperature-change stimuli). Photo-thermal conversion efficiency or thermal conductivity of the nanocomposite additive becomes the principal parameter. Thus, materials with high thermal conductivity (e.g., carbon nanotube [161], titanate nanosheet [162], and transition metal dichalchogenide [163]) are being used. For electroactive hydrogel, such as poly(2-acrylamido-2-methylpropanesulfonic acidl; PAMPS) polymer and 2-acrylamido-2-methylpropanesulfonic acid (AMPS), nanocomposites with high electric conductivity such as multi-walled CNT [164] and graphene oxide [165] can be used. These additives provide ion transport within the hydrogel, thereby generating an osmotic pressure difference which leads to swelling, de-swelling, or bending [165]. For magneto-responsive hydrogel, magnetite [166] or super paramagnetic nanoparticles [20,167] can be embedded in PAAm or PEG hydrogels. Based on the magnetic particle chain alignment design, actuation ranging from simple bending to complex swimming or crawling can be realized [168,169] (Figure 13). Additionally, under an alternating magnetic field (AMF), one could also create magneto-thermal effects [170]. It is noteworthy that, for a fast response rate, nanocomposite additives such as thermoplastic urethane [171] and titanate nanosheet [162] are the currently known best performers. Certain nanocomposites also enhance the mechanical properties of the overall hydrogel [172].

There are a number of recent reports that highlight the importance of reinventing underappreciated actuation mechanisms. A report utilized the Magangoni effect to generate programmable actuators without external stimuli [173] (Figure 14a,b). With this, the authors developed a water surface Marangoni microswimmer with high programmability. Unlike previous Marangoni microswimmers, they introduced photo-patterning to fabricate multiple functional parts and were able demonstrate time-dependent motion changes and even the disassembly of swimmer parts. They used polyvinyl alcohol (PVA) as fuel because it generates surface tension drop-based propulsion as well as has high water solubility and poor solubility in organic solvents. Another report showed a unique type of actuator that utilizes surface tension for its actuation [174] (Figure 14c,d). The authors utilized liquid surface tension to transform pen ink 2D drawings into 3D geometries and subsequently fixed it with potassium persulfate (KPS)-based surface-initiated monomer polymerization. This way, they were able to perform the large scale roll-to-roll (R2R) fabrication of 3D hydrogel objects from simple and inexpensive 2D drawings.

### 5.3. Encryption and Anti-Counterfeiting

A unique class of hydrogel-based smart materials, outside the category of sensors and actuators, is hydrogel taggant for encryption [20,175] or anti-counterfeiting purposes [176]. Unlike the conventional encoded particles we introduced above where codes are deterministic and overt, these taggants have embedded codes for covert operations sch as encryption and authentication to prevent information hacking and product counterfeiting, respectively. Counterfeiting is a serious threat to most industries. Among them, counterfeit drugs have an estimate market size of USD 431 billion [177], which not only causes a tremendous loss for pharmaceutical companies but also a health crisis in the general public. Though many authentication techniques such as holograms and RFIDs exist, even these authentication taggants are prone to hacking when the value of the product sufficiently exceeds the cost for taggant forging. Although many other materials can be utilized for anti-counterfeiting, several recent reports highlighted the virtues of harnessing hydrogels for such a purpose.

It is conceivable to deploy encoded microparticles to various industrial situations such as regarding currency, drugs, chips, and other valuable products, but this requires high encoding capacity with robust decoding in various harsh environments, such as in high temperature or the presence of harsh chemicals. A reported technique deals with this subject by using rare-earth up-conversion nanocrystals for generating encoded PEGDA particles which produces 10^6^ encoding capacity and an ultra-low decoding false-alarm rate (<10^9^). This robust and diverse encoding capability was achieved due to the narrow spectral variation (2% CV) of the nanocrystals. These particles showed robust decoding capabilities in high-temperature casting and lamination conditions, as well as high biocompatibility for miRNA detection.

When designing hydrogels with polymer additives, these additives are commonly embedded inside the hydrogel’s 3D matrix. However, in some cases, polymer additives could create new functionalities with simple changing of their deposition pattern. Bae and colleagues invented duplication-resistant human fingerprint-like wrinkled taggants based on polymer microparticles coated with silica [178] (Figure 15a,b). These wrinkled microparticles, although fabricated in batches, have individual unique wrinkle patterns with a staggering 10^135^ coding capacity. Evidently, these particles are difficult to replicate even by the original producer, thus having a fingerprint-like authentication capability when applied on products. Generating these microscale wrinkle patterns relied on the instability generated between the polymer matrix and the coated silica layer during the shrinking process. This created semi-random wrinkling patterns on the polymer microparticle surfaces. They were also able to create maze-like structures by guided wrinkling [179] (Figure 15c). Additionally, further-improved security of these fingerprints was possible with gray-scale lithography [180] (Figure 15d).

## 6. Perspective Design and Development

In light of the past successful history, we anticipate that inventions and translational research studies of novel functional hydrogels will continue. In viewing the many topics we covered, we noticed a number of trends.

First is the emergence of facile systems for real-life POC diagnostics. In the field of biosensors, we envision hydrogel-based POC diagnosis products being introduced to the market in the future. However, to meet the requirements for practical use, there are two rather intrinsic obstacles to overcome: (i) obtaining sufficient quantitative readout accuracy and (ii) increasing the shelf-life of the hydrogel device by means of sustained hydration strategies [12]. For example, DNA hydrogels have many advantages for developing facile POC diagnosis tools but further improvements to produce sufficient sensitivity with fast output quantification are still needed [32]. Alarmed by the COVID-19 pandemic, developing advanced biosensors for detecting SARS, MERS, TB, and other pathogens are re-gaining momentum. To account for the paramount need of disposable POC devices for the larger public, making the device facile and cheap became paramount. Future research on facile integration with the sample preparation module and/or readout module will further stimulate translational outputs of this field.

The second trend we observed is the invention of integrated systems for smart platform technologies. For the therapeutic field, hydrogel proved to be a great platform material across the board, from screening drugs and creating artificial tissues in vitro to delivering drugs in vivo. Developing or selecting therapeutic drugs such as antibodies and antibiotics frequently involves some type of a high-throughput screening procedure, such as the evolutionary selection assay or parallel reaction assay. To realize such high-throughput assays with reduced cost and reagent consumption, reported studies utilized hydrogel-based micro-platforms. Further generalizing these platforms will involve improving assay robustness under realistic conditions with numerous potential contaminants and larger candidate drug libraries with diverse biochemical properties. Obviously, such generalization endeavors will include iterative design optimization of the hydrogel matrices and additives. Additionally, based on our clinical experience in AST and anti-SARS-CoV-2 antibody-screening, using advanced statistical tools such as machine learning is almost indispensable when dealing with the many complexities at the clinical level. We anticipate that adapting these statistical tools to optimize either the design or the performance will become more and more prevalent for developing other smart hydrogel platforms as well.

For the molecular diagnostics field, the advent of high-throughput genomic analysis tools is allowing for precision molecular diagnostics such as single-cell genomics to enter clinic settings. Alongside this trend, hydrogels with nucleic acid or protein additives are being utilized for developing novel cell-capture platforms. Design concerns, other than good cell-capture sensitivity and selectivity, often include preserving cell viability and molecular integrity. Such cell-capture strategies that effectively sort out rare cells of interest (CTCs, stem cells, etc.) and could be coupled with highly sensitive downstream genomic analyses have great implications for both liquid biopsy-based early cancer diagnosis and many scientific research studies including stem cell, immunology, and neurology studies.

For the tissue engineering field, engineering complex organs faces challenges at probably the widest range of length scale. In any case, hydrogel structures will almost definitely be included in the architecture. We expect future inventions of smart strategies that combine the advantages of bottom-up (cell culture-based) approaches and top-down (3D cell printing-based) approaches. Tailoring a living organ naturally entails the need to incorporate dynamic and smart materials into the overall structure. One could imagine hydrogel soft robot implants performing muscle-like works, coordinated signal transduction, or the programmed assembly of natural and/or artificial cells. However, current smart material-based soft robots still suffer from their low strain, energy transduction efficiency, actuation speed, and robustness [158]. Additionally, for artificial cells, only simple biochemical or biophysical demonstrations have been introduced. However, new actuation mechanisms and design solutions are continuously being reported. As we mentioned above, there are numerous physical and chemical reactions that are still under-explored in terms of rational designing. Creating multiple compartments within artificial cells with techniques such as phase separation will help increase the number of integrated biochemical circuits. Additionally, artificial intelligence could aid in modeling and designing the dynamics among the constituents of tissues (i.e., cell-cell, cell-matrix, and matrix-matrix interaction).

One emerging trend is generating IT functionalities on hydrogels. Similar to microfabrication enabling hydrogel-based multiplexed assays, DNA writing and inorganic nanocomposites are now enabling hydrogel-based IT techniques such as denser data storage, encryption, and anti-counterfeiting. These are pioneering demonstrations opening rooms for other numerous IT techniques to be realized with such non-semiconductor settings. Apparently, virtues of using hydrogel-based IT techniques will stand out in applications where biocompatibility is an important factor. Potential fields include food, drugs, implants, and cryopreserved biosamples. These fields encompass large industries and have either direct or indirect relations with the biomedical field, thus further increasing the application spectrum and potential utility of hydrogels.

Conclusively, the biomedical utilization spectrum and potential commercial value of hydrogels are immense and continuously growing. Commercialization barriers for these hydrogel platforms are driving multidisciplinary research towards either compact/facile POC devices or complex/dynamic smart platforms depending on some factors. We expect that a careful target-oriented invention of hydrogel additive chemistries and rational multifunctional systems design will continue to drive this field.

## Figures and Tables

**Figure 1 ijms-23-02955-f001:**
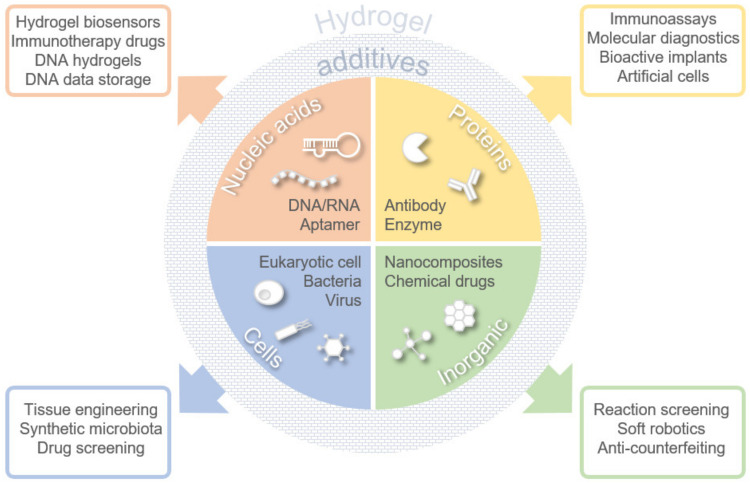
Types of hydrogel additives and their functional applications.

**Figure 2 ijms-23-02955-f002:**
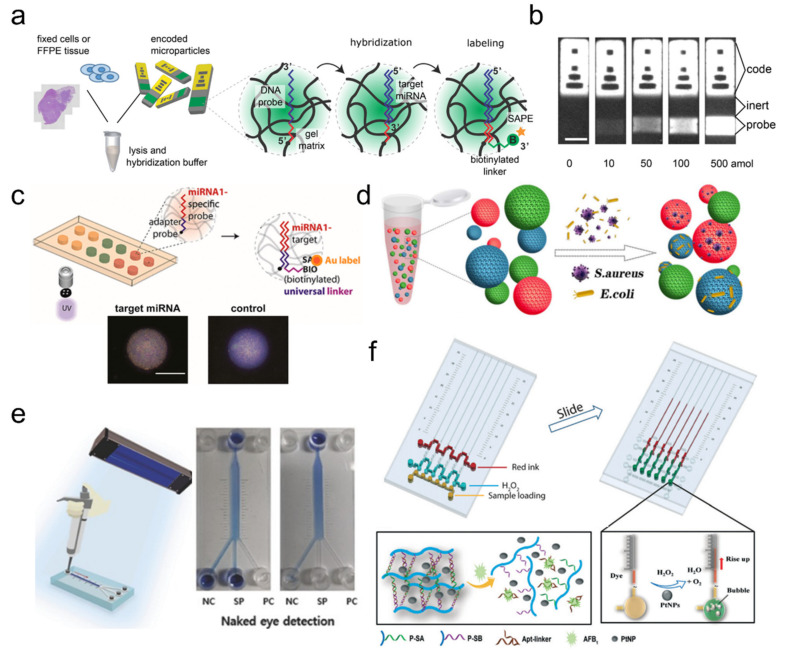
Nucleic acid-based biosensors. (**a**–**d**) Multiplex analyte detection using encoded hydrogel particles integrated with DNA probe. (**a**,**b**) Multiplexed miRNA detection using graphically encoded hydrogel microparticles; adapted with permission from [22]. Copyright 2018 ACS. (**c**) Multiplexed miRNA detection using spatially encoded hydrogel microparticles. Scale bar is 100 μm; reproduced with permission from [24]. Published under CC-BY license 2020. Copyrighted by the authors. (**d**) Multiplexed pathogen detection using colorimetrically encoded hydrogel microparticles. Microparticle colors indicate photonic reflectance barcodes; adapted with permission from [26]. Copyright 2018 Elsevier. (**e**,**f**) DNA hydrogel-based analyte detection using pure DNA hydrogel formation (**e**) or hybrid DNA hydrogel (**f**). (**e**) Rapid diagnosis of MERS pathogens using pseudo-serum sample; adapted with permission from [12]. Copyright 2016 Wiley. (**f**) Colorimetric detection of Aflotoxin B1-responsive aptamer hydrogel; adapted with permission from [27]. Copyright 2016 RSC.

**Figure 4 ijms-23-02955-f004:**
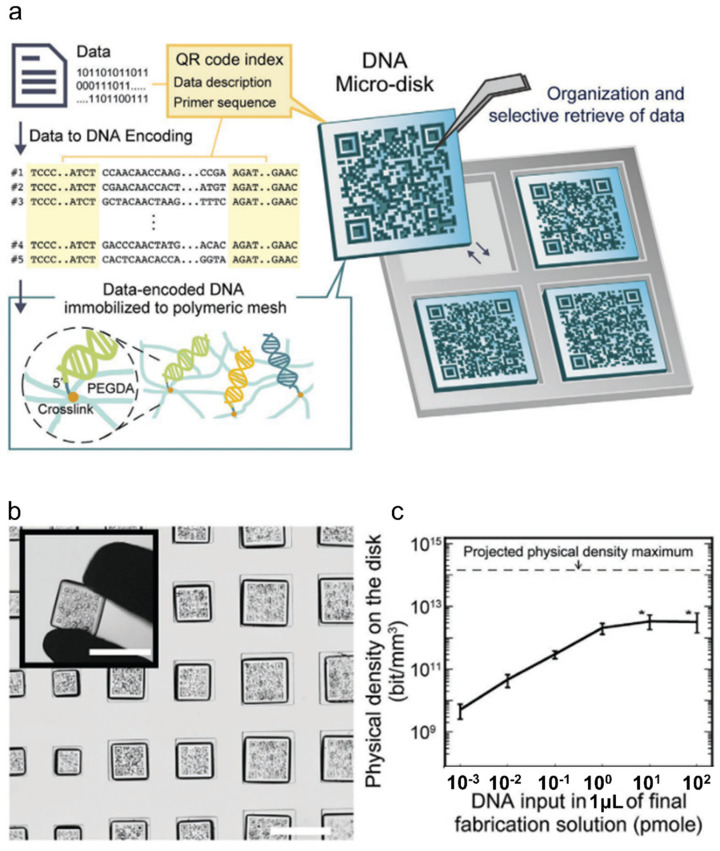
Hydrogel micro-disk-based DNA data storage. (**a**) Schematic of DNA template immobilization on hydrogel micro-disk with QR code for encoding information of data-encoded DNA. (**b**) Image of an array of DNA micro-disks; scale bar: 500 um. (**c**) Capability of tunable data storage capacity; adapted with permission from [58]. Copyright 2020 Wiley.

**Figure 5 ijms-23-02955-f005:**
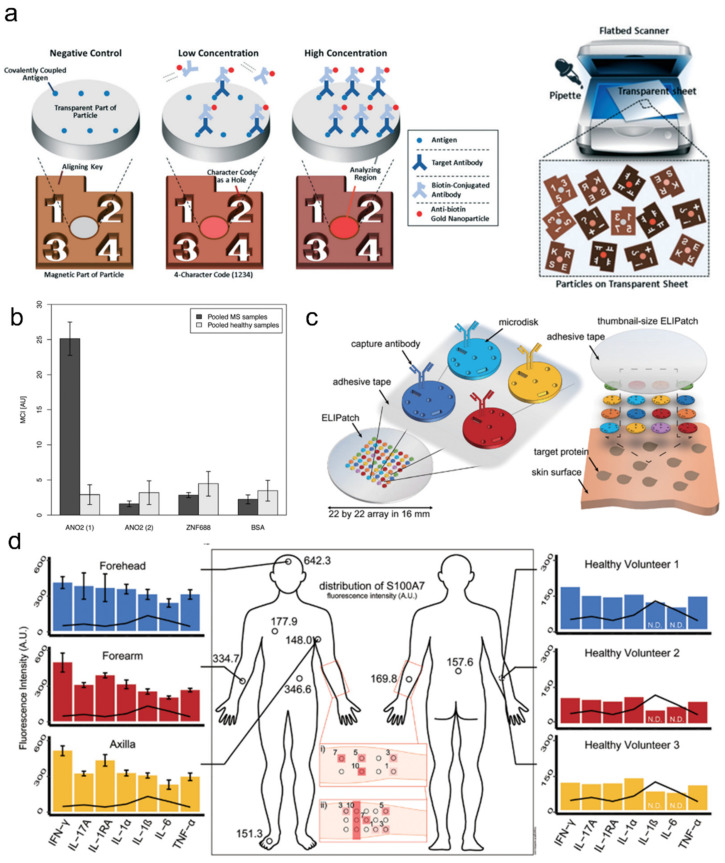
Antibody-laden hydrogel microparticles. (**a**,**b**) Barcoded antibody-laden microparticles for multiplex serum antibody detection. (**a**) The gold nanoparticle (AuNP)-based colorimetric assay was used to detect target antibody, which can be read out with conventional scanners. (**b**) Multiplex detection of autoantibodies in multiple sclerosis patient plasma samples; adapted with permission by the authors of from [67]. Copyright 2017 RSC. (**c**,**d**) Hydrogel microparticle array-loaded skin patch for multiplex protein assay. Hydrogel microparticles were incorporated with designated anti-cytokine antibodies for target cytokine detection (**c**). Skin patches containing a mixture of different barcoded microparticles were applied to various skin and subjects for multiplex skin cytokine profiling (**d**); adapted with permission from [68]. Copyright 2018 AIP.

**Figure 6 ijms-23-02955-f006:**
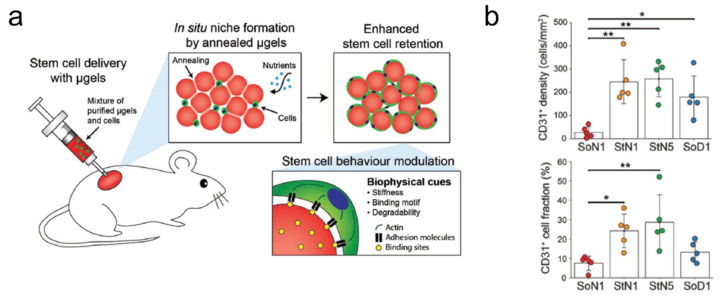
In vivo delivery of microporous annealed particles (MAPs) for wound healing. (**a**) Schematic of subcutaneous injection of MAP for effective vascular regeneration. (**b**) Results of injecting MAPs with four distinctive compositions (SoN1, StN1, StN5, and SoD1), leading to vascular endothelial cell (CD31+) regrowth in injection site; adapted with permission from [71]. Copyright 2019 Wiley.

**Figure 7 ijms-23-02955-f007:**
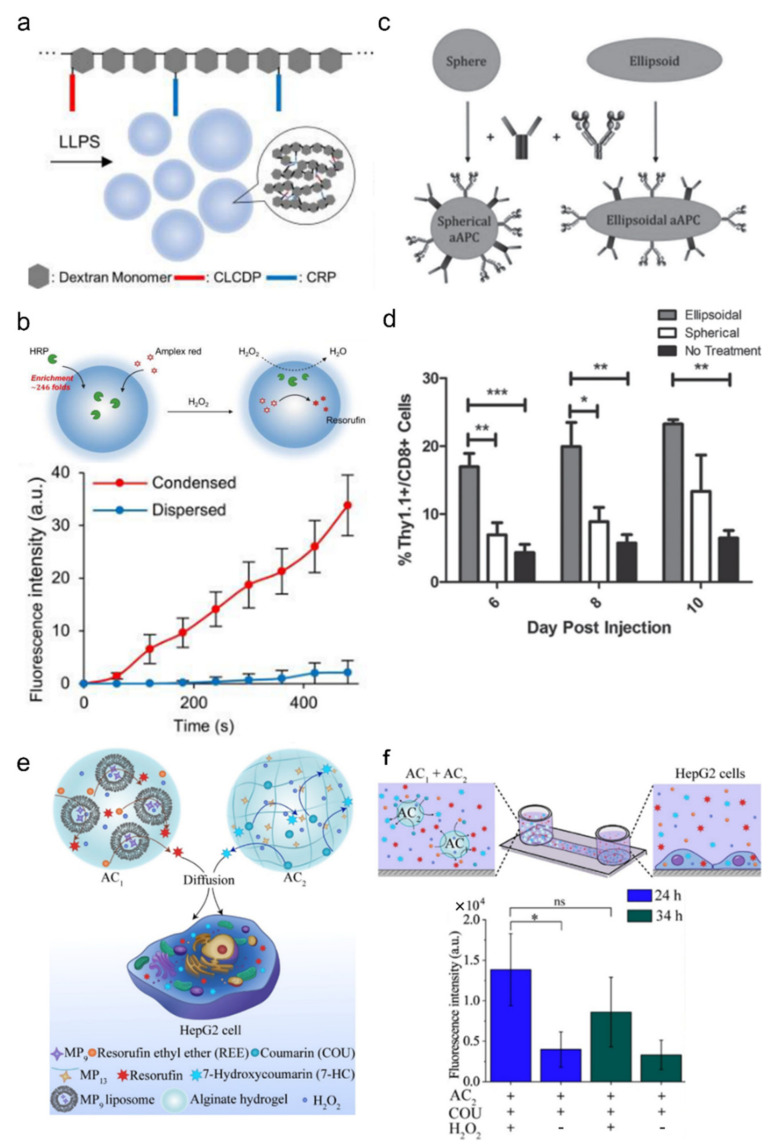
Examples of hydrogel-based artificial cells. (**a**,**b**) Liquid–liquid phase separation using polymer–peptide hybrid. (**a**) Oligopeptides were used as stickers while dextran backbones were used as spacers. These hybrid materials effectively formed membrane-less organelle-like droplets. These droplets were able to contain enzymes and carry out phase-separated reactions; adapted with permission from [94]. Published under CC-BY license (4.0). (**c**,**d**) Artificial antigen presenting cells (aAPCs) for antigen-specific T cell stimulation. (**c**) Hydrogel microparticle-based aAPCs were fabricated by incorporating anti-CD28 antibody for T cell co-stimulation and MHC Db Ig Dimer for antigen-specific stimulation. (**d**) T cells were stimulated in vivo more effectively by ellipsoidal aAPCs than spherical aAPCs; adapted with permission from [95]. Copyright 2015 Wiley. (**e**,**f**) Artificial signal transfer between artificial cells and mammalian cells. (**f**) Alginate artificial cells were used to encapsulate Coumarin and produce 7-Hydroxycoumarin, which, in turn, was transferred to HepG2 cells to induce fluorescence; adapted with permission from [96]. Copyright 2021 Wiley.

**Figure 8 ijms-23-02955-f008:**
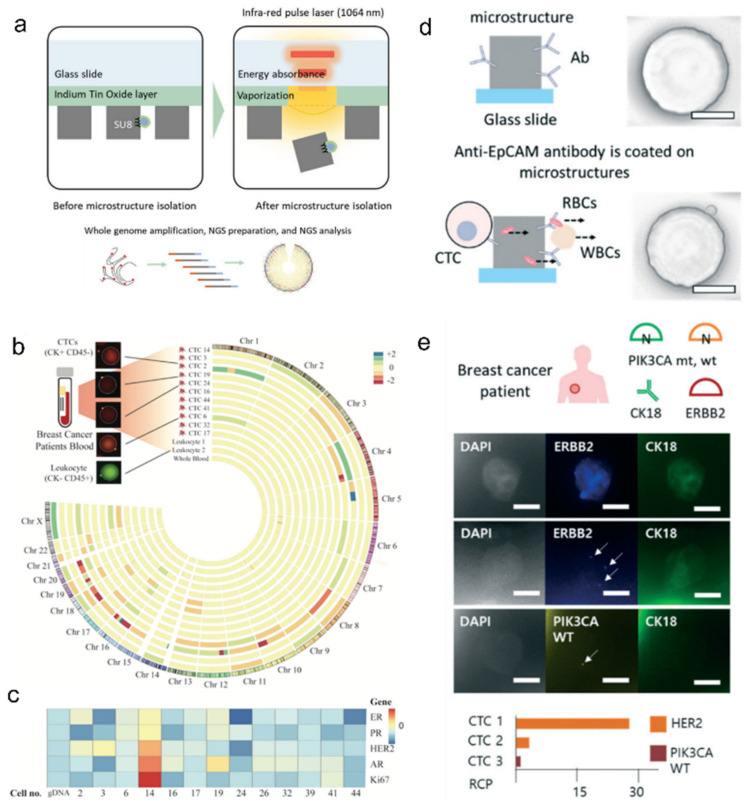
(**a**–**c**) Whole-genome sequencing of captured CTCs. Pulsed lasers were used for retrieving breast cancer patient-derived CTC. (**a**) Retrieved cells were processed for downstream whole-genome profiling (**b**) and analysis of copy number alteration of tumorigenesis-related gene loci (**c**); adapted with permission from [100]. Copyright 2019 Wiley. (**d**,**e**) Schematic of antibody-crosslinked polymer post for capturing circulating tumor cells (CTCs) from blood sample; scale bars are 50 μm. (**e**) In situ molecular profiling of CTCs from breast cancer patient; scale bars are 10μm and adapted with permission by the authors of [101]. Copyright 2020 RSC.

**Figure 9 ijms-23-02955-f009:**
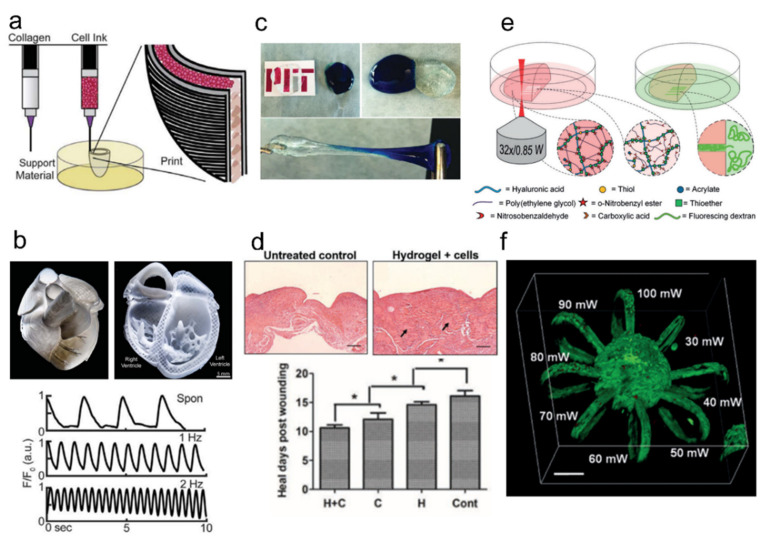
Mammalian cell-laden hydrogels for tissue engineering. (**a**,**b**) Collagen-based human heart 3D printing. (**b**) Demonstration of 3D-printed contracting ventricle and neonatal human heart printing; adapted with permission from [116]. Copyright 2019 AAAS. (**c**) Self-healing gel from PEG-phenylboronic acid and PEG-diol; adapted with permission from [119]. Copyright 2016 Wiley. (**d**) Injectable cell-laden self-healing hydrogel made of hyperbranched polyPEGDA and thiol-modified gelatin for wound repair; adapted with permission from [120]. Copyright 2017 Wiley. (**e**,**f**) Photodegradable hydrogel-based 3D cell encapsulation; adapted with permission from [121]. Published under CC-BY license 2018. (**e**) Micro-channel fabrication by two-photon degradation of PEG-HA-SH. (**f**) Demonstration of adipose-derived mesenchymal stem cells’ encapsulation and spreading in micro-loops made of PEG-HA-SH hydrogel.

**Figure 10 ijms-23-02955-f010:**
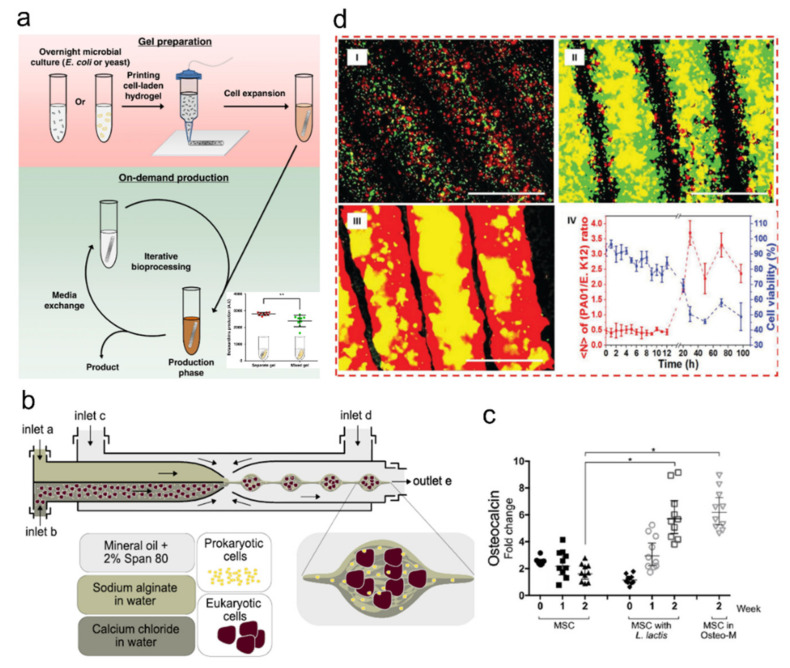
Bacteria-laden hydrogels as artificial microbiota for biomaterial production. (**a**) Hydrogel-laden microbes for on-demand production. Overview of microbe-laden extrusion-printed hydrogel. The printed and UV-cured microbial hydrogels are transferred to culture medium for outgrowth. The inset plot is the demonstration of how spatial organization of microbial consortia improves on-demand production; adapted with permission from [137]. Published under CC-BY (4.0) license 2020. (**b**,**c**) MSC differentiation in bacteria-laden hydrogels; adapted with permission from [138]. Published under CC-BY license 2019. (**b**) Method of encapsulating MSCs with differentiation protein secreting bacteria in hydrogel droplets. (**c**) Osteogenic differentiation of MSC via bacteria expressing fibronectin and BMP-2. Osteocalcin expression was higher in bacteria-laden samples as well as in samples cultured in osteogenic media than MSCs without bacteria. This means microgels enable a commensalism symbiotic relationship between bacteria and MSCs for cell differentiation. (**d**) Bacteria embedded hydrogel fiber-based microbiota model. Separate fibers contained either GFP-labeled *E. coli* K12 or RFP-labeled PA01 strains. The competitive relationship changes the viability of each strain (iv); adapted with permission from [139]. Copyright 2018 Wiley.

**Figure 11 ijms-23-02955-f011:**
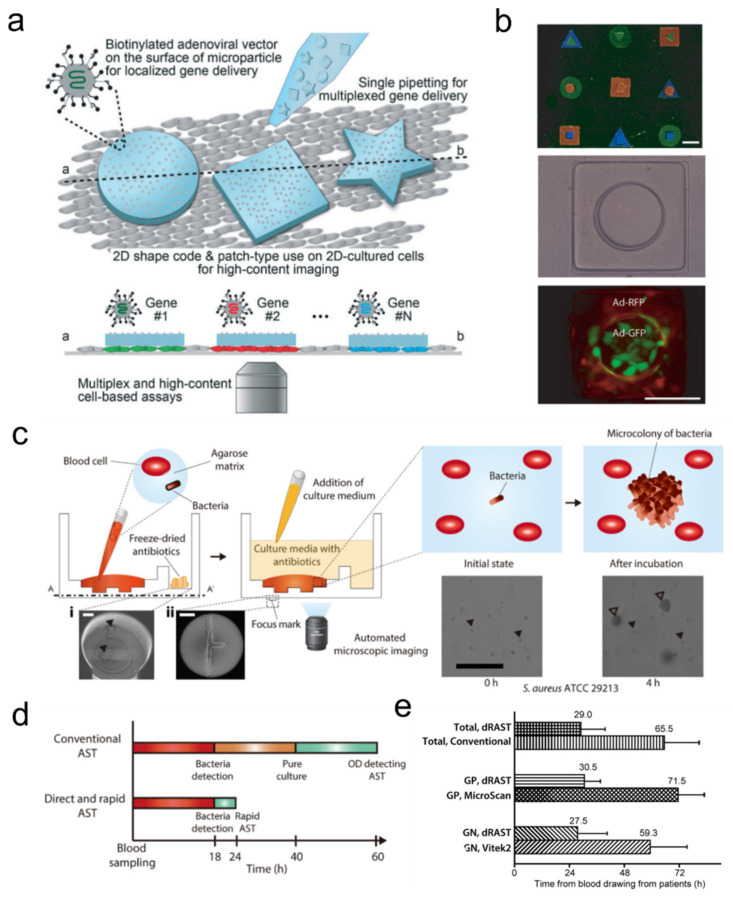
Microorganism-laden hydrogels for drug screening. (**a**) Schematic of encoded viral micropatch for multiplex cellular assays. Each shape-coded particle is conjugated with adenovirus with designated gene for localized gene transfer; adapted with permission from [142]. Copyright 2017 RSC. (**b**) Demonstration of hierarchically assembled shape-encoded hydrogel micropatch for dual gene delivery; adapted with permission from [143]. Copyright 2018 AIP (**c**–**e**). Direct detection of antibiotic resistance from positive blood culture bottle by image tracing of agarose-embedded bacteria cells; adapted with permission from [144]. Published under CC-BY license 2017. (**c**,**d**) Illustration of direct blood detection of single-bacteria microcolony by antibiotic-laden agarose hydrogel-embedding of blood. (**e**) Comparison of test duration between conventional AST and direct AST.

**Figure 12 ijms-23-02955-f012:**
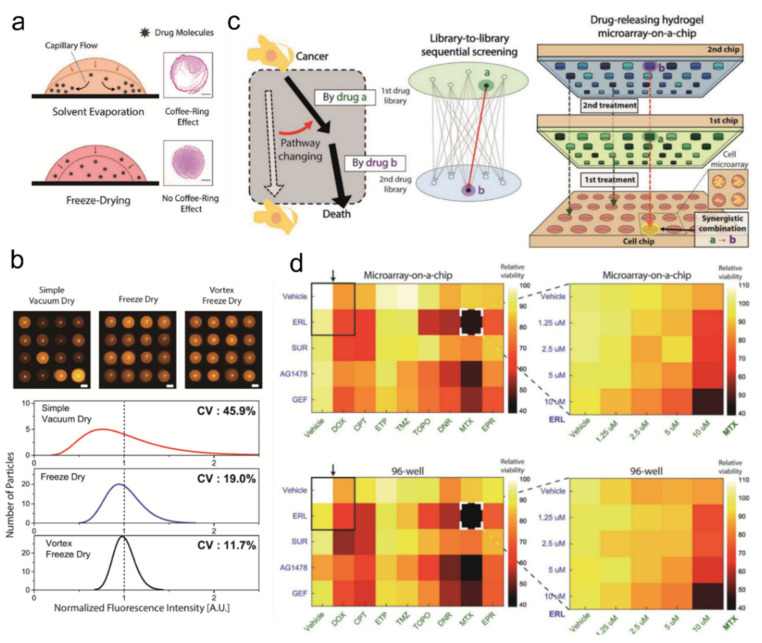
Chemical drug-laden hydrogel microparticles for drug screening. (**a**,**b**) Uniform drug loading into prefabricated microparticles. (**a**) Comparison of drying uniformity between solvent evaporation-based drying and freeze drying. (**b**) Demonstration of fluorescent rhodamine-B drying on hydrogel microparticles with different drying methods. Coefficient of variation was used for measuring drying uniformity; adapted with permission from [156]. Copyright 2017 Wiley. (**c**,**d**) Schematic (**c**) and demonstration (**d**). Drug-releasing hydrogel microarray for high-throughput drug combination screening. (**d**) Sequential anti-cancer drug combination assay against BT-20 cancer cell line. The result shows good agreement between conventional 96-well experiment and microarray chip-based method; adapted with permission from [157]. Published under CC-BY license 2018.

**Figure 13 ijms-23-02955-f013:**
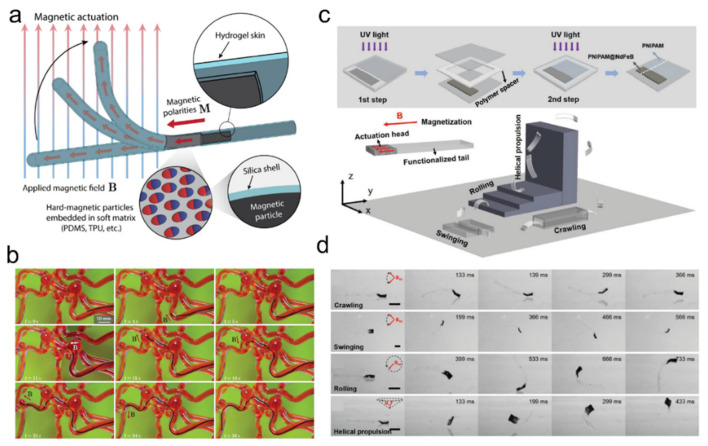
Magnetic particle-laden hydrogels as soft robots. (**a**,**b**) Schematic (**a**) and demonstration (**b**) of hydrogel-coated ferromagnetic polymer soft robots; adapted with permission from [168]. Copyright 2019 AAAS. (**c**,**d**) Schematic (**c**) and locomotion demonstration (**d**) of soft untethered millirobot (iRobot). iRobot is capable of crawling, swinging, rolling, and helical propulsion in water at room temperature using oscillating magnetic field (~10 mT); adapted with permission from [169]. Copyright 2020 Wiley.

**Figure 14 ijms-23-02955-f014:**
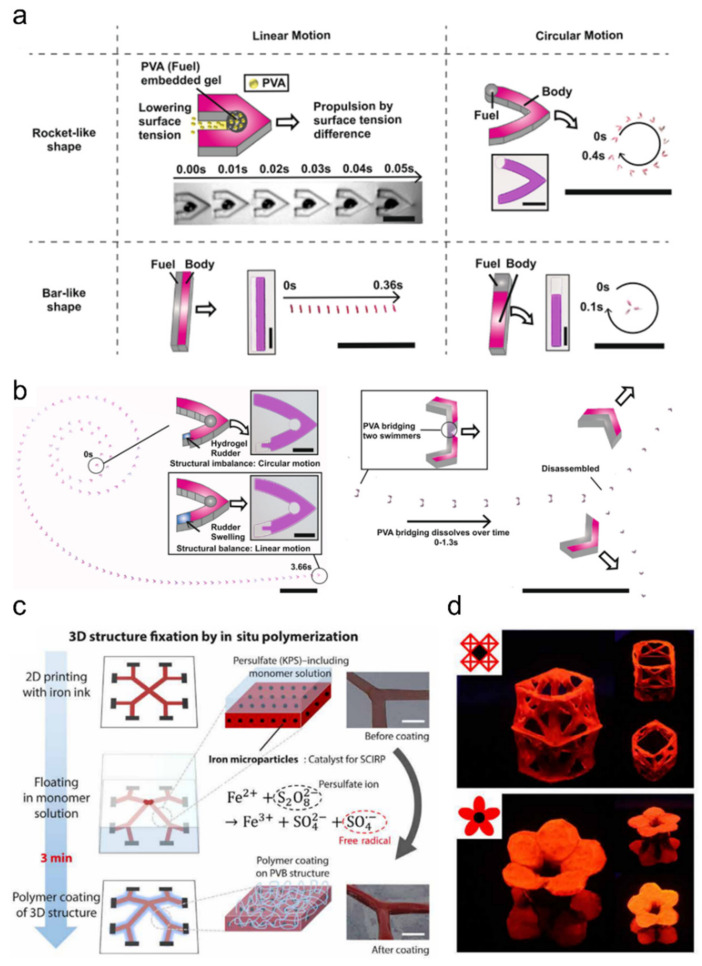
(**a**,**b**) Schematic and demonstration of programmable Marangoni microswimmers made of PVA-embedded hydrogel. Embedded PVA is used as fuel, which reduces the local surface tension, leading to propulsion without external stimuli; adapted with permission from [173]. Copyright 2021 NPG. Published under CC-BY license 2021. (**c**,**d**) Schematic (**c**) and demonstration (**d**) of 3D hydrogel transformation of 2D pen drawing. After surface tension-based floating of the 2D pen-drawn PVB film, embedded iron microparticles induce local polymerization surrounding the PVB film; adapted with permission by the authors of [174]. Copyright 2021 AAAS.

**Figure 15 ijms-23-02955-f015:**
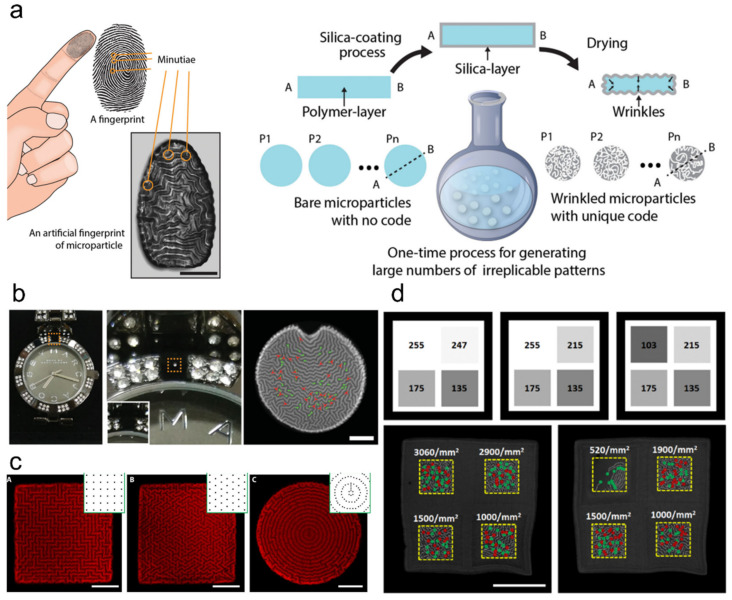
(**a**,**b**) Schematic (**a**) and demonstration (**b**) of human fingerprint-like microparticle-based product anti-counterfeit tagging. Drying of thin silica layer-coated PEGDA microparticles induces random surface wrinkle generation, which is different for every particle. These unique winkling patterns can be utilized as authentication taggants; scale bar is 100 μm; adapted with permission from [178]. Copyright 2015 Wiley. (**c**) Demonstration of hydrogel microparticles with physical mazes based on guided wrinkle patterning; scale bars are 25 μm; adapted with permission by the authors of [179]. Copyright 2017 AAAS. (**d**) Demonstration of gradient wrinkling for higher security level anti-counterfeiting; scale bar is 100 μm; adapted with permission by the authors of [180]. Published under CC-BY-NC-ND license.

## Data Availability

Not applicable.

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
