# Peer review of "Recent Advances in Polymer Additive Engineering for Diagnostic and Therapeutic Hydrogels"

_ijms, 2022, doi:10.3390/ijms23062955_

Round 1

Reviewer 1 Report

The manuscript provides a wealth of knowledge on the use of (even somewhat controversial) hydrogel materials in unique forms of application (i.e. intelligent materials, soft robotics, and DNA data storage), and therefore makes an outstanding contribution to the field of hydrogel systems. The manuscript is written in an appropriate language, it is informative and understandable, and the graphics presented correspond to the subject matter discussed. However, before publication, I suggest introducing a few minor modifications:

- the abstract could be modified as it does not show the full scientific value of the peer-reviewed work in its current form;

- authors should consider changing the title of the last paragraph to "Perspective design and development";

- the font in the drawings needs to be enlarged because it is difficult to read;

- some editing is needed, e.g. line 485 wound up instead of not, missing punctuation marks etc..

Author Response

To the reviewer, 

Thank you for your positive review and detailed comments. The following is our response to each of our comments. 

  • response to comment 1: We added a few sentences to show our full scope of "translational research of functional hydrogels" such as those mentioning i) the types of diagnostic/therapeutic applications, ii) the general trends like POC and integrated smart platforms, and iii) future perspective.
  • response to comment 2: We changed the title according to your suggestion.
  • response to comment 3: We made our best effort to make the fonts as large as possible. Please excuse us that due to our limited editing capabilities we just enlarged the respective plots or images to make the text bigger.
  • response to comment 4: Thank you for your keen observation. We screened the manuscript and corrected any other punctuation errors we could find. 

The attached is the revised manuscript.

Thank you. 

Reviewer 2 Report

In this manuscript entitled “Recent advances in polymer additive engineering for diagnostic and therapeutic hydrogels”, the authors summarized recent studies of hydrogels for biomedical application and other engineering applications. Overall, this is an informative and useful review. This review will attract the attention of many researchers in biology, chemistry, and medicine. In my opinion, this paper is fit for the journal scope of “International Journal of Molecular Sciences”. So, I recommend publication. However, I have a minor comment in line 499.

Page 11, line 419: “library of 5 × 105 complexity” should be written as “library of 5 × 105 complexity”.

Author Response

To the reviewer, 

Thank you for your positive review. We will make the correction as you pointed out. 

The attached file is the revised manuscript. 

Thank you.
